# BENCHMARKING AGENTIC WORKFLOW GENERATION

**Shuofei Qiao**[♠][*], **Runnan Fang**[♠][*], **Zhisong Qiu**[♠][*], **Xiaobin Wang**[◇],
**Ningyu Zhang**[♠][†], **Yong Jiang**[◇][†], **Pengjun Xie**[◇], **Fei Huang**[◇], **Huajun Chen**[♠][♡][†]
[♠]Zhejiang University  [◇]Alibaba Group
[♡]Zhejiang Key Laboratory of Big Data Intelligent Computing
{shuofei,zhangningyu}@zju.edu.cn

## ABSTRACT

Large Language Models (LLMs), with their exceptional ability to handle a wide range of tasks, have driven significant advancements in tackling reasoning and planning tasks, wherein decomposing complex problems into executable workflows is a crucial step in this process. Existing workflow evaluation frameworks either focus solely on holistic performance or suffer from limitations such as restricted scenario coverage, simplistic workflow structures, and lax evaluation standards. To this end, we introduce WORFBENCH, a unified workflow generation benchmark with multi-faceted scenarios and intricate graph workflow structures. Additionally, we present WORFEVAL, a systemic evaluation protocol utilizing subsequence and subgraph matching algorithms to accurately quantify the LLM agent's workflow generation capabilities. Through comprehensive evaluations across different types of LLMs, we discover distinct gaps between the sequence planning capabilities and graph planning capabilities of LLM agents, with even GPT-4 exhibiting a gap of around 15%. We also train two open-source models and evaluate their generalization abilities on held-out tasks. Furthermore, we observe that the generated workflows can enhance downstream tasks, enabling them to achieve superior performance with less time during inference[1].

*"If you can't describe what you are doing as a process, you don't know what you're doing."*

— W. Edwards Deming

## 1 INTRODUCTION

The remarkable advances in Large Language Models (LLMs) (Dubey et al., 2024; Yang et al., 2024a; OpenAI, 2024) are gradually recovering the considerable potential of LLM-driven agents towards tackling complex real-world problems, such as function calls (Qin et al., 2024; Tang et al., 2023; Qu et al., 2024; Liu et al., 2024a), embodied planning (Song et al., 2023; Zeng et al., 2024a; Xiang et al., 2023; Song et al., 2024; Qiao et al., 2024a), code generation (Hong et al., 2024; Qian et al., 2023; Zhang et al., 2024c), etc., wherein decomposing complex problems into executable-granularity subtasks is a crucial capability that LLM agents must possess to achieve practically deployable.

A set of subtasks with execution dependencies is typically referred to as a **workflow**. Workflows can serve as an intermediate state for solving complex tasks, aiding agents in bridging the gap between tasks and specific executable actions (Wang et al., 2024). Moreover, an explicit workflow can enhance the agent's debuggability and interpretability, facilitating human-machine interaction. Many previous studies have examined the potential of LLMs for automatic workflow generation (Zeng et al., 2023; Ye et al., 2023; Xue et al., 2024; Li et al., 2024c). Recent researchers in the realm of LLM agents emphasize workflows as a form of prior knowledge or experience, guiding agent planning to avoid the occurrence of hallucinations (Zhu et al., 2024; Xiao et al., 2024; Wang et al., 2024). As for benchmarks in this field, most works are confined to the end-to-end planning

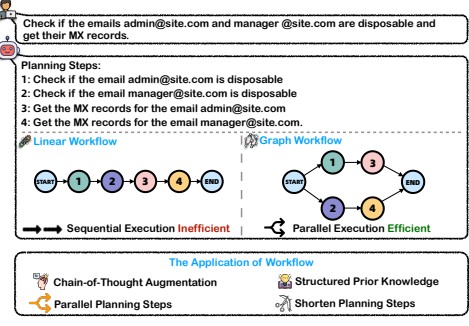

Figure 1: **Workflow and its application.**

---

[*] Equal Contribution.
[†] Corresponding Author.
[1]Code and dataset are available at https://github.com/zjunlp/WorfBench.

ability of LLM agents (Qin et al., 2024; Patil et al., 2023; Yang et al., 2023b; Wu et al., 2024; Liu et al., 2024c), while only a few studies attempt to evaluate the workflow generation (problem decomposition) capability from a finer granularity perspective (Shen et al., 2023; Chen et al., 2024b; Ye et al., 2024; Zheng et al., 2024a; Valmeekam et al., 2023). However, the current evaluation benchmarks unavoidably suffer from the following issues: 1) **Limited scope of scenarios**. They only focus on function calling or reasoning tasks. 2) **Sole emphasis on linear relationships between subtasks** (Figure 1 left). Real-world scenarios often involve more complex graph structures, including parallelism. 3) **Evaluations heavily rely on GPT-3.5/4**, yet LLM themselves exhibit hallucinations and ambiguity. These limitations make the evaluation of workflow generation lack some systematicity. Therefore, to address the above issues, we introduce WORFBENCH, a unified workflow generation benchmark with the following features:

- **Multi-faceted scenarios.** We cover four complex scenarios for LLM agents, including problem-solving, function calling, embodied planning, and open-grounded planning. The dataset comprises 18k training samples, 2146 test samples, and 723 held-out tasks to evaluate generalization.

- **Complex workflow structures.** We model the workflows as Directed Acyclic Graphs based on dependencies between subtasks, enabling a more precise representation of complicated serial or parallel structures in the real world (Figure 1 right).

- **Strict quality control and data filtering.** We introduce an intermediary structure called node chain between the original task and the workflow graph and employ the Topological Sorting algorithm for rigorous validation of the graph structure followed by human evaluation.

- **Accurate quantitative evaluation.** We further propose WORFEVAL to evaluate the workflow generation ability of LLM agents, which applies developed subsequence and subgraph matching algorithms for accurate quantitative assessment.

We evaluate the performance of most mainstream LLMs with various model scales on WORFBENCH. We observe that, compared to linear structures, the models' ability to predict graph-structured workflows falls far short of real-world requirements, with even GPT-4 (OpenAI, 2023) only achieving a performance of 52.47%. Furthermore, we train two open-sourced models on the training set and evaluate their generalization abilities on held-out tasks. Finally, we analyze how workflows can enhance end-to-end model performance as CoT (Wei et al., 2022) augmentation and prior knowledge, reducing end-to-end inference time by paralleling and shortening planning steps (Figure 1 bottom).

To sum up, we summarize our main contributions as follows:

- We propose WORFBENCH, a unified workflow generation benchmark with multi-faceted scenarios and complex workflow structures. We conduct strict data filtering and human evaluation to ensure the quality of WORFBENCH.

- We introduce WORFEVAL, using effective subsequence and subgraph matching algorithms to evaluate the workflow generation ability of LLM agents from both chain and graph structures.

- We conduct comprehensive evaluation on various closed-sourced and open-sourced models with different scales. We further exploit the generated workflows to facilitate downstream tasks and achieve superior and efficient performance.

## 2 WORFBENCH

### 2.1 TASK FORMULATION

Given a specific task and a candidate action list, our goal is to enable the language agents to generate a graph-structured workflow, where the nodes in the workflow satisfy the minimum executable granularity. Our action list here can include function APIs, tools, embodied actions, or mixture of the above to simultaneously adapt various scenarios. Formally, given the task description $q$, action list $\mathcal{A}$ and language agents $\mathcal{M}_\theta$, the workflow generation can be modeled as:

$$\mathcal{G}(\mathcal{V}, \mathcal{E}) \leftarrow \mathcal{M}_\theta(q, \mathcal{A}), \quad (1)$$

where $\mathcal{G}$ is a Directed Acyclic Graph (DAG) with nodes $\mathcal{V} = \{v_1, v_2, ..., v_{|\mathcal{V}|}\}$ being subtasks and edges $\mathcal{E} = \{(v_i, v_j)\}, 1 \leq i \neq j \leq n$ representing the execution relationships between nodes

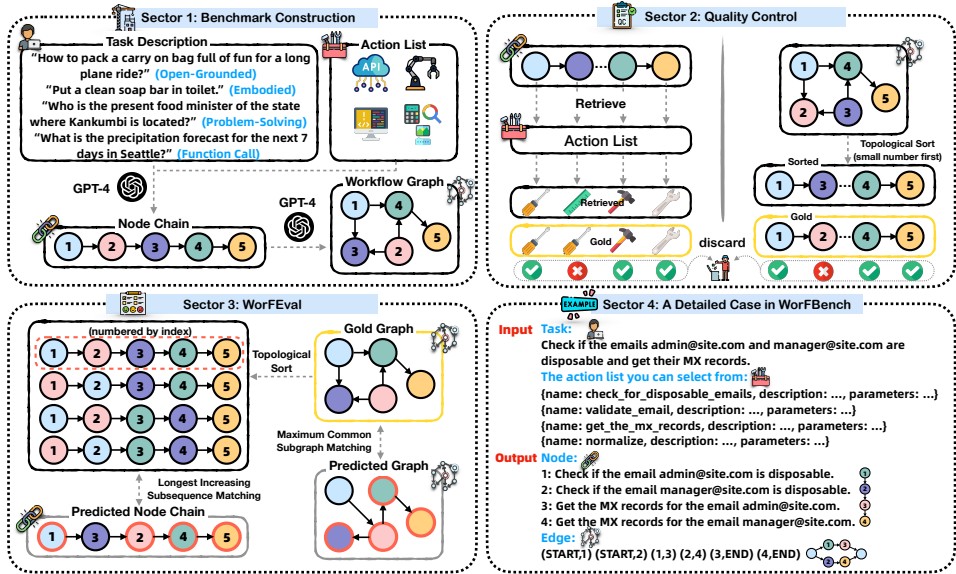

Figure 2: **The overview framework of our WORFBENCH. Sector 1** is the benchmark construction where we first synthesize the node chain and then the workflow graph (§2.2). **Sector 2** is our data filtering process (§2.3). **Sector 3** describes the algorithms in WORFEVAL to evaluate the predicted workflow of LLM agents (§2.4). **Sector 4** is a detailed data point of our WORFBENCH. Note that each node in this figure is uniquely identified by its color. Numbers on the nodes represent their indexes in the gold chain. Nodes matched with gold chain or graph are circled by ⭕ in Sector 3.

($v_j$ must be executed after $v_i$). **Note that all nodes $\mathcal{V}$ need to be executed according to their dependencies before task $q$ is considered completed.** It is evidently a bit challenging to directly instruct the language agents to generate a graph structure. To accommodate the generation habits of language models, we introduce the node chain $\mathcal{C}(\mathcal{V})$ before graph $\mathcal{G}(\mathcal{V}, \mathcal{E})$:

$$\mathcal{G}(\mathcal{V}, \mathcal{E}) \leftarrow \mathcal{C}(\mathcal{V}) \leftarrow \mathcal{M}_\theta(q, \mathcal{A}). \quad (2)$$

Here node chain $\mathcal{C}$ is one of the Topological Sequences of graph $\mathcal{G}$, where the order of nodes in the node chain ensures the relative order of nodes in the graph. Thus, the workflow graph generation can be reformulated by first creating a node chain and then establishing edges for the generated nodes.

## 2.2 BENCHMARK CONSTRUCTION

We mainly collect various tasks $q$ and the corresponding action lists $\mathcal{A}$ from existing well-known datasets. To facilitate a better understanding of the benchmark construction, we provide a detailed exposition of each dataset utilized in our paper in Appendix A.1. To ensure the quality of the data, we also adhere to the strategy of first constructing the node chain and then building the workflow graph during benchmark construction. Depending on the varied ways of acquiring workflow nodes in different scenarios, we categorize node chain $\mathcal{C}$ construction into the three following types:

**Function Call Tasks.** Our function call data is collected from ToolBench (Qin et al., 2024) and ToolAlpaca (Tang et al., 2023), both of which involve combining different functions to accomplish multi-step user tasks. As the two datasets only have golden function calls for each task, to make the synthesized node chain more reasonable, following the *thought-action-observation* REACT (Yao et al., 2023) format, we first utilize GPT-4 to reverse-engineer the *thought* based on the given function call (*action*) and then execute the function call to obtain the *observation*. Next, we carefully design the few-shot prompt for GPT-4 to generate the node (subtask) for each step based on its *thought-action-observation* loop. By iterating through all the function call steps, we can obtain the node chain for the specific task. In addition, we also include Seal-Tools (Wu et al., 2024) as the held-out tasks, where the construction of the node chain follows the same process as described above.

**Embodied Tasks.** We collect the REACT format gold trajectories of ALFWorld (Shridhar et al., 2021) and WebShop (Yao et al., 2022) from ETO (Song et al., 2024) and OS (Liu et al., 2024b) from AgentInstruct (Zeng et al., 2024a). Unlike function call tasks, where each function call corresponds to a node, embodied scenarios evolve dynamically based on the environment. It is hard to decompose tasks into a one-node-per-action granularity solely based on the initial environment. Therefore, we can only decompose tasks into a fixed granularity based on the task and initial environmental information available. However, the advantage of embodied tasks is that the workflows of the same kind of tasks are similar. So for each kind of task, we seriously analyze and manually design few-shot

examples, enabling GPT-4 to synthesize node chains directly based on the gold trajectories. Similarly, we also include InterCodeSQL (Yang et al., 2023a) as held-out tasks for embodied scenarios.

🎯**Problem-Solving and** 🎨**Open-Grounded Tasks.** We also introduce problem-solving tasks like math, commonsense, and multimodal reasoning tasks from LUMOS (Yin et al., 2024), as well as a challenging general open-grounded dataset, WikiHow (Koupaee & Wang, 2018). Since the LUMOS-O version of LUMOS and the WikiHow have already contained gold planning chains, we directly process them into the unified format we require. Furthermore, since WikiHow does not provide candidate action lists, we use the specific task as the query to retrieve the most similar actions in the public action library as distractors and then mix the distractors with the gold actions to create the action list. The aim of retrieving similar actions is to increase the difficulty of the task.

After obtaining the node chain $\mathcal{C}(\mathcal{V})$, we further use GPT-4 to generate edges for the node chain. Since each node is a piece of text describing a subtask, it is formally difficult to add edges to the text directly. Thus, we sequentially assign numbers to the nodes in the node chain and use these numbers instead of the node text when generating edges (e.g. $(i, j)$ instead of $(v_i, v_j)$). All the generated edges finally form $\mathcal{G}(\mathcal{V}, \mathcal{E})$. For ease of evaluation, we add the START and END nodes to represent the beginning and end of a workflow. A detailed data point is illustrated in Figure 2 Sector 4. All the detailed prompts we use during the benchmark construction process can be found in Appendix A.7.

## 2.3 QUALITY CONTROL

Although we use GPT-4 to synthesize workflows, rationality still cannot be guaranteed. So we conduct restricted quality controls and data filtering for both node chain and workflow graph generation.

**Quality Control for Node Chain.** Due to the low variability of the node chains in embodied tasks, our primary focus lies in filtering the node chain of function call tasks. Two critical factors are whether the order of nodes is logical and whether each node accurately decomposes the task. The former has been guaranteed when constructing the node chain based on the sequence of gold function calls. For the latter, we use each synthesized node as a query to retrieve the function list. If one node within the node chain retrieves a function that does not align with the gold, we will discard this task. We filter out 15.36% data through the quality control for the node chain.

**Quality Control for Workflow Graph.** The key priority is to ensure that the order of nodes in the workflow graph is consistent with the order of nodes in the node chain. Therefore, we perform a Topological Sort on the graph generated by GPT-4. Since the topological sort of a graph is not unique, to ensure uniqueness and facilitate matching with the node chain, during the topological sorting process, when there are multiple nodes with an in-degree of $0$ in the graph, we sequentially remove them in ascending order based on their node number defined in node chain. We discard the data points with topological sorting results not aligned with node chains. We filter out 29.77% data through the quality control for the workflow graph.

To ensure complexity, we filter out data points with only $1$ node or $1$ edge. Furthermore, to maintain data balance across different scenarios, we randomly sample datasets with excessive data volume, subsequently dividing them into training and testing sets. Then we manually check the test set for a fair and effective evaluation (The detailed **human verification process** can be found in Appendix A.2). The final data statistics are outlined in Appendix A.3.

## 2.4 WORFEVAL

To ensure accuracy, apart from utilizing GPT-4 or simple semantic similarity matching, we quantitatively evaluate both the node chain and workflow graph using restrict algorithms. Assuming the nodes and edges of the gold workflow are denoted as $\mathcal{V}^g = [v_1^g, v_2^g, ..., v_{|\mathcal{V}^g|}^g]$ and $\mathcal{E}^g = [e_1^g, e_2^g, ..., e_{|\mathcal{E}^g|}^g]$, while the agent predicted nodes and edges are represented as $\mathcal{V}^p = [v_1^p, v_2^p, ..., v_{|\mathcal{V}^p|}^p]$ and $\mathcal{E}^p = [e_1^p, e_2^p, ..., e_{|\mathcal{E}^p|}^p]$. Firstly, we calculate the similarity matrix $\mathcal{S}$ between $\mathcal{V}^g$ and $\mathcal{V}^p$:

$$\mathcal{S}_{i,j} = \begin{cases} \sigma(v_i^g, v_j^p), & \sigma(v_i^g, v_j^p) \geq \beta \\ 0, & \sigma(v_i^g, v_j^p) < \beta \end{cases} \tag{3}$$

Here, $1 \leq i \leq |\mathcal{V}^g|, 1 \leq j \leq |\mathcal{V}^p|$. $\sigma$ represents the cosine similarity function and in this paper, we encode the semantics of nodes using Sentence-BERT[2] (Reimers & Gurevych, 2019). $\beta$ serves as a

---

[2]`all-mpnet-base-v2`: `https://huggingface.co/sentence-transformers/all-mpnet-base-v2`. This model is also used as the retriever in benchmark construction process.

threshold, and we consider two nodes to be semantically matched only when their similarity is greater than or equal to $\beta$. Therefore, $\mathcal{S}$ can be viewed as a bipartite graph, where one part consists of gold nodes and the other part consists of predicted nodes. Since a predicted node may match multiple gold nodes and a gold node may be matched by multiple predicted nodes, we utilize a **max-weighted bipartite matching algorithm** (Hopcroft & Karp, 1973) to find the best matches. Ultimately, the matched gold nodes and predicted nodes form two new node sets $\mathcal{V}^{g'} \subseteq \mathcal{V}^g$ and $\mathcal{V}^{p'} \subseteq \mathcal{V}^p$, where the nodes in $\mathcal{V}^{p'}$ and the nodes in $\mathcal{V}^{g'}$ correspond one-to-one. After obtaining the aforementioned variables, we proceed to evaluate both the node chain and workflow graph predicted by the agent.

**Node Chain.** Supposing the agent's predicted node chain is $\mathcal{C}(\mathcal{V}^p)$, based on the gold workflow graph $\mathcal{G}(\mathcal{V}^g, \mathcal{E}^g)$, we can obtain all its possible topological sequences $\{\mathcal{C}(\mathcal{V}^g)_1, \mathcal{C}(\mathcal{V}^g)_2, ..., \mathcal{C}(\mathcal{V}^g)_n\}$[3]. Assuming $\mathcal{C}(\mathcal{V}^{p'})$ is a permutation of $\mathcal{V}^{p'}$ such that it forms a subsequence of $\mathcal{C}(\mathcal{V}^p)$, and considering the one-to-one mapping between $\mathcal{V}^{p'}$ and $\mathcal{V}^{g'}$, we can derive a sequence $\mathcal{C}(\mathcal{V}^{g'})$ that aligns in order with $\mathcal{C}(\mathcal{V}^{p'})$. Next, following Chen et al. (2024b), based on the indexes of $\mathcal{V}^{g'}$ in $\mathcal{C}(\mathcal{V}^g)_i, 1 \leq i \leq n$, we can calculate a **Longest Increasing Subsequence (LIS)**[4] $l_i$ of $\mathcal{C}(\mathcal{V}^{g'})$ for each $\mathcal{C}(\mathcal{V}^g)_i$:

$$l_i = \text{LIS}(\mathcal{C}(\mathcal{V}^{g'}), \mathcal{C}(\mathcal{V}^g)_i) \tag{4}$$

Then, $l = \max(|l_1|, |l_2|, ..., |l_n|)$ is obtained to represent the length of the longest valid subsequence for the agent's predicted node chain $\mathcal{C}(\mathcal{V}^p)$. Finally, the score for the node chain is denoted as:

$$f1_{\text{chain}} = \frac{2p_{\text{chain}}r_{\text{chain}}}{p_{\text{chain}} + r_{\text{chain}}}, p_{\text{chain}} = \frac{l}{|\mathcal{V}^p|}, r_{\text{chain}} = \frac{l}{|\mathcal{V}^g|}, \tag{5}$$

where $p_{\text{chain}}$ and $r_{\text{chain}}$ are the precision and recall of the generated node chain respectively.

**Workflow Graph.** Based on $\mathcal{V}^{p'}$, we can derive the subgraph $\mathcal{G}(\mathcal{V}^{p'}, \mathcal{E}^{p'})$, where $\mathcal{E}^{p'}$ satisfies $\mathcal{E}^{p'} \subseteq \mathcal{E}^p$ and $\forall (v_i, v_j) \in \mathcal{E}^{p'}, v_i, v_j \in \mathcal{V}^{p'}$. By once again leveraging the one-to-one correspondence between $\mathcal{V}^{p'}$ and $\mathcal{V}^{g'}$, we can utilize the **Maximum Common Induced Subgraph (MCIS)**[5] matching algorithm to find the maximum match between $\mathcal{G}(\mathcal{V}^{p'}, \mathcal{E}^{p'})$ and $\mathcal{G}(\mathcal{V}^g, \mathcal{E}^g)$:

$$\mathcal{G}_{mcis}(\mathcal{V}_{mcis}, \mathcal{E}_{mcis}) = \text{MCIS}(\mathcal{G}(\mathcal{V}^{p'}, \mathcal{E}^{p'}), \mathcal{G}(\mathcal{V}^g, \mathcal{E}^g)) \tag{6}$$

Assuming the number of nodes in this maximum common induced subgraph is $k = |\mathcal{V}_{mcis}|$, the score for the agent-generated workflow graph is denoted as:

$$f1_{\text{graph}} = \frac{2p_{\text{graph}}r_{\text{graph}}}{p_{\text{graph}} + r_{\text{graph}}}, p_{\text{graph}} = \frac{k}{|\mathcal{V}^p|}, r_{\text{graph}} = \frac{k}{|\mathcal{V}^g|}, \tag{7}$$

where $p_{\text{graph}}$ and $r_{\text{graph}}$ are the precision and recall of the generated workflow graph respectively.

# 3 EXPERIMENTS

## 3.1 EXPERIMENTAL SETTINGS

To comprehensively evaluate the workflow generation capabilities of existing LLM agents, we validate a total number of 18 models on the test set of WORFBENCH, including: **1)** Four representative **closed-sourced** LMs: O1 (`o1-preview`) (OpenAI, 2024), GPT-4 (`gpt-4-turbo-2024-04-09`) (OpenAI, 2023), GPT-3.5 (`gpt-3.5-turbo-0125`) (OpenAI, 2022), and Claude-3.5 (`claude-3.5-sonnet-1022`) (Anthropic, 2024). **2)** Fifteen state-of-the-art **open-sourced** LMs ranging from 7B to 72B: Llama series models and their variants, Llama-3.1-{8,70}B (`Meta-Llama-3.1-{8,70}B-Instruct`) (Dubey et al., 2024), Llama-2-13B (`Llama-2-13b-chat-hf`) (Touvron et al., 2023), Vicuna-13B (`vicuna-13b-v1.5`) (Zheng et al., 2023), and WizardLM-{13,70}B (`WizardLM-13B-V1.2` and `WizardLM-70B-V1.0`) (Xu et al., 2024); Qwen series models, Qwen-2-{7,72}B (`Qwen2-{7,72}B-Instruct`)

---

[3]When the number of nodes and edges is considerable, matching all possible topological orders of a graph becomes a high time complexity issue. Therefore, we limit the output to a maximum of 20 topological orders. For a specific analysis of why it is 20, please refer to Appendix A.9.

[4]https://en.wikipedia.org/wiki/Longest_increasing_subsequence

[5]https://en.wikipedia.org/wiki/Maximum_common_induced_subgraph

Table 1: **Main Results.** We evaluate all the models with identical carefully designed instructions and two-shot examples. We categorize the models based on whether the models are open-source and their scales. The best results for each category are marked in **bold**, and the second-best results are marked with underline.

| Model | 🧰 Function Call | | ✏️ Problem-Solving | | 🏆 Embodied | | 🧭 Open-Grounded | | Average | |
|---|---|---|---|---|---|---|---|---|---|---|
| | $f1_{\text{chain}}$ | $f1_{\text{graph}}$ | $f1_{\text{chain}}$ | $f1_{\text{graph}}$ | $f1_{\text{chain}}$ | $f1_{\text{graph}}$ | $f1_{\text{chain}}$ | $f1_{\text{graph}}$ | $f1_{\text{chain}}$ | $f1_{\text{graph}}$ |
| *Closed-Sourced* | | | | | | | | | | |
| Claude-3.5 | 66.44 | 55.06 | 67.28 | 55.50 | **71.74** | **56.71** | **61.33** | **42.88** | 66.70 | **52.53** |
| GPT-3.5 | 73.36 | 60.32 | 69.86 | 54.50 | 64.57 | 49.17 | 47.67 | 28.10 | 63.86 | 48.02 |
| GPT-4 | **74.87** | **62.11** | 67.18 | 55.24 | 70.94 | 56.17 | 56.30 | 36.36 | **67.32** | 52.47 |
| O1-preview | 70.68 | 57.11 | **72.76** | **59.25** | 69.90 | 54.19 | 53.47 | 35.97 | 66.70 | 51.63 |
| *Open-Sourced* | | | | | | | | | | |
| GLM-4-9B | 59.27 | 36.34 | 58.91 | 40.15 | 53.17 | 36.15 | 44.04 | 22.56 | 53.85 | 33.80 |
| Phi-3-small | 57.66 | 40.71 | 55.76 | 39.75 | 54.77 | 37.52 | **44.65** | 22.66 | 53.21 | 35.16 |
| Llama-3.1-8B | 63.30 | 43.62 | 64.49 | 46.79 | 56.23 | 36.40 | 44.58 | **25.48** | 57.15 | 38.08 |
| Mistral-7B | 67.30 | 51.67 | 61.27 | 45.35 | 64.59 | **48.83** | 40.97 | 21.48 | 58.53 | 41.83 |
| Qwen-2-7B | **70.79** | **55.50** | 68.65 | 52.13 | 62.83 | 46.25 | 39.29 | 20.89 | 60.39 | 43.69 |
| InternLM-2.5-7B | 68.43 | 52.99 | 72.92 | 57.80 | 65.77 | 48.09 | 40.84 | 21.27 | 61.99 | 45.03 |
| Llama-2-13B | 53.32 | 34.33 | 53.74 | 38.69 | 44.24 | 30.55 | 37.17 | 23.14 | 47.12 | 31.68 |
| WizardLM-13B | 55.78 | 36.94 | 65.42 | 49.71 | 55.41 | 37.34 | 37.23 | 21.66 | 53.46 | 36.41 |
| Vicuna-13B | 53.75 | 37.66 | 64.58 | 50.25 | 57.99 | 42.61 | 38.93 | 23.11 | 53.81 | 38.41 |
| Qwen-1.5-14B | 65.73 | 46.86 | 58.80 | 43.89 | 60.55 | 44.14 | 41.73 | 21.44 | 56.70 | 39.08 |
| Phi-3-medium | 67.71 | 47.26 | **71.15** | 54.85 | 65.11 | 49.99 | 42.73 | 23.77 | 61.68 | 43.97 |
| WizardLM-70B | 63.47 | 45.46 | 63.92 | 47.93 | 59.15 | 42.87 | 45.27 | 26.89 | 57.95 | 40.79 |
| Mixtral-8×7B | 66.13 | 48.83 | **71.89** | 57.58 | 72.08 | 54.94 | 42.96 | 23.21 | 63.26 | 46.14 |
| Llama-3.1-70B | 64.41 | **52.72** | 70.37 | 57.05 | 69.98 | 55.52 | **53.64** | **33.06** | 64.60 | 49.59 |
| Qwen-2-72B | **71.67** | 52.31 | 70.63 | **58.13** | 73.24 | 58.49 | 53.43 | 32.89 | 67.24 | 50.46 |

(Yang et al., 2024a) and Qwen-1.5-14B (`Qwen1.5-14B-Chat`) (Bai et al., 2023); Mistral series models, Mistral-7B (`Mistral-7B-Instruct-v0.2`) (Jiang et al., 2023) and Mixtral-8×7B (`Mixtral-8x7B-Instruct-v0.1`) (Jiang et al., 2024); Phi series models, Phi-3-{small, medium} (`Phi-3-{small,medium}-128k-instruct`) (Abdin et al., 2024); other models including GLM-4-9B (`glm-4-9b-chat`) (Zeng et al., 2024b) and InternLM-2.5-7B (`internlm2.5-7b-chat`) (Cai et al., 2024).

We evaluate all the models using the LlamaFactory (Zheng et al., 2024b) framework with two-shot prompting. For models above 70B parameters, we utilize vLLM (Kwon et al., 2023) to accelerate inference. The hyperparameters used during decoding are all set to default values except for the temperature, which is 0.5. For all the models, the semantically matching threshold $\beta$ is set to 0.6.

## 3.2 MAIN RESULTS

Table 1 displays our detailed experimental results. With the help of this table, we aim to analyze the following four research questions.

**Q1: Which is more challenging for LLM agents, linear planning or graph planning?** For each kind of task, we show both the node chain score $f1_{\text{chain}}$ and workflow graph score $f1_{\text{graph}}$, representing the linear planning and graph planning abilities of LLM agents respectively. It can be observed that the workflow graph scores of all models are significantly lower than the node chain scores, with the largest disparity found in GLM-4-9B, reaching an average difference of 20.05%. Even the model with the smallest difference, Llama-3.1-70B, exhibits a difference of 15.01%. We also explore what will happen if we provide the gold node chains to the agents and task it solely with predicting the edges of the workflow graph (in Appendix A.4 Table 5). Despite the noticeable improvement after being provided with node chains, the model's performance in graph planning is still unsatisfactory, aligning with our analysis in Q4 Figure 4. So it appears that graph planning is more challenging than linear planning, and overall, the graph planning capability of models is positively correlated with their linear planning capability. However, there are also several exceptions to this trend. For instance, in the reasoning tasks, Qwen-2-72B has a lower $f1_{\text{chain}}$ compared to Mixtral-8×7B, but it leads in $f1_{\text{graph}}$. On the other hand, in function call tasks, although Qwen-2-72B has a more satisfying $f1_{\text{chain}}$, its $f1_{\text{graph}}$ is lower than that of Llama-3.1-70B.

**Q2: How is Scaling Law manifested in workflow generation?** Model size, training data scale, and training time are three crucial indicators of the Scaling Law (Kaplan et al., 2020). For open-source models, we categorize them into three groups based on model size: around 7B, 13B, and 70B. By observing the performance of models within the same series, we can discern the power of model size for workflow generation. For instance, in the Qwen-2 series, the 72B model outperforms the 7B model by 6.77% in $f1_{\text{graph}}$; in the Llama-3.1 series, the 70B model surpasses the 8B model by 11.51%. An unconventional phenomenon is that some 7B models outperform the majority of 13B models. One reason for this is that most of the selected 7B models were released within the past six months, and trained on a greater amount of high-quality data, which is crucial for workflow

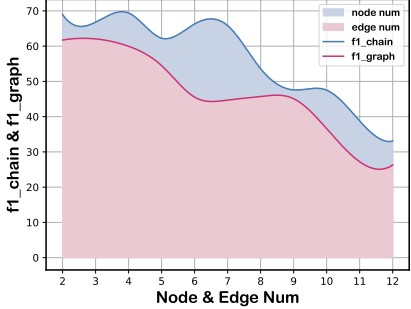

Figure 3: **Performance Distribution of GPT-4**. The distribution of $f1_{\mathrm{chain}}$ for the number of nodes and the distribution of $f1_{\mathrm{graph}}$ for the number of edges.

Table 2: **Generalization Results** of fine-tuned (FT) models on held-out tasks compared to baselines.

| Model | Held-in Tasks | | Held-out Tasks | | | |
| --- | --- | --- | --- | --- | --- | --- |
| | Average | | 🔧Seal-Tools | | 🏆InterCodeSQL | |
| | $f_{\mathrm{chain}}$ | $f_{\mathrm{graph}}$ | $f_{\mathrm{chain}}$ | $f_{\mathrm{graph}}$ | $f_{\mathrm{chain}}$ | $f_{\mathrm{graph}}$ |
| GPT-3.5 | 63.86 | 48.02 | 95.91 | 76.63 | 65.30 | 53.07 |
| GPT-4 | 67.32 | 52.47 | **96.58** | 80.25 | **66.35** | **54.36** |
| Qwen-2-7B | 60.39 | 43.69 | 92.68 | 74.75 | 54.20 | 39.72 |
| InternLM-2.5-7B | 61.99 | 45.03 | 93.07 | 74.43 | 55.06 | 42.20 |
| Phi-3-medium | 61.68 | 43.97 | 94.11 | 79.45 | 58.45 | 46.62 |
| Llama-3.1-70B | 64.60 | 49.59 | 94.40 | 80.11 | 63.49 | 53.66 |
| Qwen-2-72B | 67.24 | 50.46 | 94.47 | 78.90 | 63.86 | 52.47 |
| Qwen-2-7B+FT | **79.35** | **70.38** | 96.49 | 82.82 | 62.37 | 48.72 |
| InternLM-2.5-7B+FT | 78.98 | 69.33 | 95.83 | **83.72** | 63.78 | 50.97 |

generation. In contrast, the 13B models were mostly released last year, and their training data and techniques may have become outdated. Another possible reason is our speculation that models around 13B may have reached a point where the trade-off between training costs and model effectiveness has become challenging to navigate. This could also explain why there have been few models of this scale released in the past six months in the open-source community.

**Q3: How far are existing LLM agents from being real workflow planning experts?** Even though the node chains and workflow graphs are synthesized by GPT-4 in our benchmark, after we remove the gold trajectories and let it generate directly, its absolute performance in $f1_{\mathrm{chain}}$ and $f1_{\mathrm{graph}}$ averages only 67.32% and 52.47%, respectively. In the challenging open-grounded tasks, the top performer, Claude-3.5, only achieves 61.33% and 42.88%, which is far from the level of a practical and deployable workflow planner. The current most powerful reasoning model O1 (OpenAI, 2024) only performs relatively well on problem-solving (reasoning) tasks, falling short on other tasks that require more environmental knowledge. In addition, we analyze the performance of GPT-4 across different numbers of nodes and edges in workflow, as shown in Figure 3. With the increase of nodes and edges, both the $f1_{\mathrm{chain}}$ and $f1_{\mathrm{graph}}$ performance of GPT-4 tend to decline, with occasional brief spikes likely caused by uneven sample distribution. Therefore, for complex planning tasks with more planning steps, the performance of GPT-4 is unsatisfying no matter for linear planning or graph planning, let alone other models. This is clearly inadequate for many complex real-world scenarios, which is why many agent architectures are currently only at the theoretical level.

Furthermore, we attempt to train the 7B models Qwen-2-7B and InternLM-2.5-7B (both models show excellent performance in Table 1) on the training set[6]. We evaluate the trained models' capabilities on both held-in and held-out tasks. Results are presented in Table 2. While the trained models have shown the best performance on Seal-Tools (surpassing GPT-4 by 2.5~3.5% in $f1_{\mathrm{graph}}$), their advantages are not as pronounced as on held-in tasks (where they surpass GPT-4 by 10%+ and 15%+ in $f1_{\mathrm{chain}}$ and $f1_{\mathrm{graph}}$, respectively). Moreover, the workflows of Seal-Tools are relatively simple (about 2~3 nodes per task on average), with even untrained 7B models achieving around 74%. When it comes to the more complex InterCodeSQL tasks, the trained models fall slightly behind, only outperforming models around 7B and 13B. To sum up, although the trained models have made significant breakthroughs on the held-in tasks, their performance does not exhibit remarkable generalization when extended to held-out tasks, especially on embodied tasks. This implies that the structured workflow planning capability cannot be learned solely through fitting a large amount of data. So whether through prompting or training for generalization, LLM agents still have a long distance to reach real workflow planning experts.

**Q4: What shall we do to enhance the workflow generation capability of LLM agents?** First, we answer this question by analyzing samples where GPT-4 scores less than 0.5 on $f1_{\mathrm{graph}}$. Through meticulous manual checks and categorization, we identify four kinds of typical errors: 1) **Granularity.** The decomposition of subtasks does not meet the minimum executable granularity. 2) **Explicitness.** The summary of subtasks is overly vague. 3) **Graph.** The subtask is correct, but the graph structure is incorrect. 4) **Format.** The output does not adhere to the specified text format. Figure 4 displays the probabilities of different types of errors. We also show some cases of O1 in Appendix A.6. Among them, case (a) represents the error of Graph, while case (b) represents the error of Granularity. We summarize that most of these errors can likely be

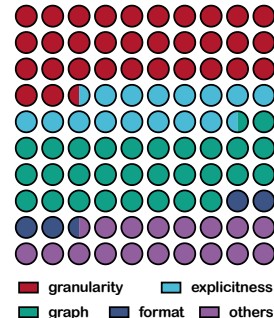

Figure 4: **Error Statistics**.

---

[6]The detailed training setups can be found in Appendix A.5

attributed to the agent's lack of environmental knowledge. Granularity pertains to a deficiency in prior knowledge of the environment. For instance, in case (b), O1 lacks the knowledge that *the fridge can be used for cooling*, assuming that a cool potato can only be achieved inside a fridge rather than somewhere else. Explicitness reflects a lack of understanding of the environmental task, resulting in insufficiently specific subtasks. Graph issues arise from a lack of comprehension regarding the dependencies of environmental actions, leading to errors in the relationships between subtasks. Some prompt optimization algorithms (Yüksekgönül et al., 2024; Zhou et al., 2024) or multi-agent architectures (Zhuge et al., 2024; Zhou et al., 2023; Hong et al., 2024) may lead to improvements to some extent. However, to achieve a higher level of intelligence, the key may lie in truly integrating world knowledge (Yu et al., 2024; Qiao et al., 2024a; Guetta et al., 2024) or world model (Dawid & LeCun, 2023; Hu & Shu, 2023; Wong et al., 2023) into agents to advance their understanding of the real world (Sumers et al., 2024; Durante et al., 2024a; Yang et al., 2024d).

# 4 THE ROLE OF WORKFLOW FOR AGENT PLANNING

In this section, we delve into how structured workflows can aid downstream tasks such as function invocation or embodied planning in achieving more precise and expedited results. In this section, we default to utilizing the trained Qwen-2-7B (mentioned in §3.2 Q3) as the workflow generator.

## 4.1 ENHANCE END-TO-END PERFORMANCE

Table 3: **End-to-end Performance** augmented by workflow as prior knowledge.

| Model | ALFWorld | | WebShop |
| --- | --- | --- | --- |
| | seen | unseen | |
| GPT-4 | 27.14 | 28.36 | 55.62 |
| GPT-4+W | **40.71** ↑13.57 | **47.01** ↑18.65 | **56.49** ↑0.87 |
| Llama-3.1-8B | 1.49 | 5.00 | 51.03 |
| Llama-3.1-8B+W | **8.21** ↑6.72 | **12.14** ↑7.14 | **52.28** ↑1.25 |
| Qwen-2-72B | 53.57 | 56.72 | 58.95 |
| Qwen-2-72B+W | **56.43** ↑2.86 | **62.29** ↑5.57 | **60.55** ↑1.60 |

**Workflow as Structured Prior Knowledge.** Given that a workflow encompasses a detailed execution process for a task, an evident use case is to employ it as prior knowledge to directly guide agent planning. This is particularly advantageous in embodied scenarios where LLM agents often lack prior knowledge of the real environment and rely on brainless trial-and-error (Qiao et al., 2024a). Therefore, we directly input the generated workflow along with the task and design instructions for the LLM agent to plan based on the guidance of the workflow. We choose GPT-4, Llama-3.1-8B, and Qwen-2-72B as the LLM agents and report the results in Table 3, illustrating that models with varying capabilities can benefit when enriched with structured workflow knowledge. For ALFWorld with greater diversity in environmental changes and more complex tasks, workflow knowledge yields greater advantages. Furthermore, we observe that these workflows are generated by a 7B model, providing guidance even to the significantly more powerful 72B model. This leads us to contemplate the weak-guide-strong paradigm, wherein a small model possessing specific environmental knowledge supervises the planning of a larger, more general model.

**Workflow as CoT Augmentation.** Chain-of-Thought (CoT) (Wei et al., 2022) has been widely acknowledged for enhancing the reasoning abilities of LLMs and plays a crucial role in OpenAI's latest reasoning model, o1 (OpenAI, 2024). However, a tricky issue lies in its long-context nature, which may mislead LLM agents in making erroneous decisions, especially when there are multiple planning steps involved. Based on our workflow construction process where each node corresponds to a function call, we can leverage this characteristic to induce agents to engage in more focused planning. Specifically, we prompt Qwen-2-7B to generate a CoT at each step based on the corresponding node and then use the node as a query to retrieve the most similar API from the API list as the func-

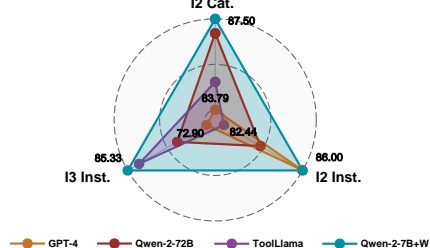

Figure 5: **Relative Function Call Accuracy** of workflow-augmented Qwen-2-7B (Qwen-2-7B+W) on StableToolBench (Guo et al., 2024b) compared with various baselines.

tion for that step. Ultimately, we allow the model to decide how to invoke the function based on the CoT and the selected function. In this process, the workflow plays a role similar to augmenting CoT, assisting the agent in thinking at each step, serving as the query for retrieval to provide the agent with more relevant APIs, thereby alleviating the agent's burden and enabling it to focus more on how to invoke tools effectively. By comparing the accuracy of function calls with ToolLlama

(Qin et al., 2024) and two one-shot baselines (Qwen-2-72B and GPT-4) on StableToolBench [7] , we find that the above procedure is effective (shown in Figure 5). Unlike a kind of external knowledge for reference, the workflow here actively participates in the planning process, leading to improved accuracy in function invocation.

## 4.2 REDUCE END-TO-END INFERENCE-TIME

**Parallel Planning Steps.** In a graph-structured workflow, nodes without dependencies can be executed in parallel. This can significantly reduce the time required to complete tasks compared to linear step-by-step execution. Continuing our analysis on StableToolbench, for a specific task, we calculate the time taken by ToolLlama to complete each node when executing step by step (including generating thought, generating function calls, executing functions, and returning results). We then mark the nodes in the workflow graph based on their completion times. So our objective can be transferred to identify the longest path between the START and END nodes, also known as the **Critical Path** of the graph. Finally, we compare the average time taken to complete all tasks with the linear ToolLlama, as shown in Figure 6. It can be observed that with graph-structured parallelization, there is a significant reduction in the average time to complete tasks, with reductions ranging from approximately one-fifth to one-third across different test sets. The parallelization feature of graph structures allows for substantial savings in inference time in real-world applications. Moreover, the execution of a node does not necessarily depend on all previous nodes, which to some extent alleviates the issue of long contexts in multi-step complex tasks, thereby enhancing the quality of task completion.

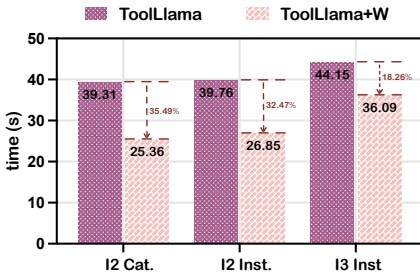

Figure 6: **Average Task Execution Time** of linear ToolLlama and parallel ToolLlama.

Table 4: **Average Planning Steps**.

| Model | ALFWorld | | WebShop |
| | seen | unseen | |
| --- | --- | --- | --- |
| GPT-4 | 17.19 | 17.43 | 5.80 |
| GPT-4+W | **15.64** ↓1.55 | **15.85** ↓1.58 | **5.72** ↓0.08 |
| Llama-3.1-8B | 19.81 | 19.43 | 7.08 |
| Llama-3.1-8B+W | **19.09** ↓0.72 | **18.38** ↓1.05 | **6.77** ↓0.31 |
| Qwen-2-72B | 14.39 | 14.67 | 3.88 |
| Qwen-2-72B+W | **14.05** ↓0.34 | **13.94** ↓0.73 | **3.73** ↓0.15 |

**Shorten Planning Steps.** In addition to the horizontal reduction of inference time brought by parallel subtask execution, we also observe that workflows can vertically decrease the planning steps of the LLM agent. This finding emerges during our experiments on workflow as structured knowledge. When the LLM agent lacks prior knowledge of the environment, it often accumulates knowledge through random trial-and-error in the environment, which may introduce irrelevant noise and lead to a drop in long-text disaster. Introducing knowledge makes the agent's actions more purposeful, reducing the steps of blind trial-and-error. In Table 4, we quantitatively analyze the average planning steps required for the model to complete tasks with or without workflow knowledge, which corroborates our discoveries.

## 5 RELATED WORK

### 5.1 LARGE LANGUAGE AGENTS

The rise of Large Language Models (LLMs) has established them at the forefront of the quest for Artificial General Intelligence (AGI), offering substantial support for the advancement of LLM-centered AI agents (Wang et al., 2023; Xi et al., 2023; Guo et al., 2024a; Yang et al., 2024b; Durante et al., 2024b; Li et al., 2024a; Zhang et al., 2024a; Yang et al., 2024c). Numerous efforts have been dedicated to employing language agents for tool utilization (Qin et al., 2024; Tang et al., 2023; Yuan et al., 2024; Ye et al., 2024; Qiao et al., 2024b; Chen et al., 2024a; Qu et al., 2024), embodied planning (Huang et al., 2022; Yao et al., 2023; Song et al., 2023; Xiang et al., 2023; Palo et al., 2023), software engineering (Hong et al., 2024; Qian et al., 2023; 2024), etc. However, these agent

---

[7]As defined in ToolBench (Qin et al., 2024), **I2** and **I3** stand for intra-category and intra-collection multi-tool instructions, respectively, based on the tools belonging to the same RapidAPI (https://rapidapi.com/hub) category or collection. **Inst.** represents unseen instructions for the same set of tools in the training data. **Cat.** denotes unseen tools that belong to an unseen category of tools in the training data.

methods or frameworks tend to concentrate primarily on the end-to-end performance of the task at hand, overlooking the assessment of the inherent reasoning and planning capabilities that are actually the bases for achieving stable and reliable agent performance.

## 5.2 WORKFLOW AND LANGUAGE AGENT PLANNING

Outside the context of LLMs, there have been various mature works in the field of concurrency theory that theoretically study the formal models for concurrent execution, including Petri nets (Peterson, 1977; Reisig, 1985; Murata, 1989) and process calculi (Milner, 1989; Hoare, 1985). Building upon these concepts and methods, much previous literature concentrates on workflow modeling and mining within the realm of business process management and process mining (van der Aalst et al., 2003a; van der Aalst, 1997; Dijkman et al., 2008; van der Aalst et al., 2003b; van der Aalst, 2016). Recently, more focus has been placed on integrating workflows with LLM agents to automate the generation of workflows or to enhance the capabilities of LLMs in handling complex problems (Wang et al., 2024; Zhang et al., 2024b; Li et al., 2024c; Xiao et al., 2024; Zeng et al., 2023). **On the one hand**, workflows can serve as an intermediate state for solving complex tasks, aiding agents in bridging the gap between tasks and specific executable actions (Li et al., 2024c; Zhang et al., 2024b). Explicit workflows can enhance the interpretability of the LLM agent, facilitating human involvement in debugging and ensuring the agent's security (human-machine interaction). **On the other hand**, workflows can serve as structured prior knowledge, assisting agents in handling knowledge-intensive tasks to avoid planning hallucinations (Zeng et al., 2023; Ye et al., 2023; Zhu et al., 2024; Qiao et al., 2024a; Xiao et al., 2024; Wang et al., 2024). Note that the DAG-based workflow introduced in our paper is equivalent in expressivity to Marked Graphs (Commoner et al., 1971), one of the many restricted subclasses of Petri nets.

## 5.3 AGENTIC WORKFLOW GENERATION AND EVALUATION

The most straightforward approach is human-designed workflows that constrain the planning process of language agents in the form of natural language prompts or state machines to prevent hallucinations (Hong et al., 2024; Li et al., 2024b; Guan et al., 2024; Zhu et al., 2024; Wang et al., 2024). However, manual design is time-consuming and lacks flexibility. As a result, some studies (Zhou et al., 2023; Ye et al., 2023; Zeng et al., 2023; Li et al., 2024c; Xue et al., 2024; Zhang et al., 2024b) try to enable language agents to automatically generate workflows. For example, Zhang et al. (2024b) use MCTS (Monte Carlo Tree Search) to guide LLM in exploring the workflow space, which effectively advances the quality of the generated workflows. So it is crucial to effectively evaluate the quality of workflows generated by language agents. Previous studies have explored automating workflow generation evaluation in tool learning or reasoning scenarios (Chen et al., 2024b; Ye et al., 2024; Shen et al., 2023; Valmeekam et al., 2023), using fine-grained metrics to assess each stage of tool utilization. In particular, problem decomposition ability is evaluated through semantic similarity matching or GPT-4 scoring. However, these works suffer from the following limitations: **firstly**, most of them focus only on function-calling or reasoning scenarios, neglecting embodied interactive scenarios with the real environment; **secondly**, their workflows are all single linear structures, making it difficult to represent more complex tasks; **lastly**, they mostly rely on GPT-4 or human evaluation. Moreover, Xiao et al. (2024) also explore workflow-guided planning, but it is noticed that they mainly examine the workflow compliance capability of language agents. Lal et al. (2024) examine the understanding of node dependencies in workflows by LLM in a question-answering format. The above two can be seen as the downstream procedures of our work.

## 6 CONCLUSION

In this work, we introduce WORFBENCH, a unified agentic workflow generation benchmark with miscellaneous scenarios and intricate graph-structured workflows. To precisely assess the workflow generation capability of LLM agents, we further present WORFEVAL, which utilizes quantitative algorithms to evaluate both the linear and graph workflows. Through comprehensive experiments across various kinds of LLMs, we investigate large performance gaps between the traditional linear-structured and complex graph-structured workflow generation. We also train open-sourced models and evaluate their generalization abilities on held-out tasks. Finally, we explore the role of workflows for downstream planning tasks from model performance and inference time efficiency.

## REPRODUCIBILITY STATEMENT

We have submitted all the test datasets (comprising both held-in and held-out tasks) of our benchmark, along with the test code, to the supplementary materials. The detailed benchmark construction and quality control processes can be found in Section 2.2 and 2.3. Additional data source information, human verification processes, and benchmark statistics are provided in Appendix A.1, A.2 and A.3. Specific test configurations: the code framework used, the model versions used, and the inference hyperparameters are mentioned in Section 3.1. All the training settings involved in our paper are detailed in Appendix A.5.

## ETHICS STATEMENT

This study is carried out in strict accordance with ethical guidelines and best practices in research. All the data utilized are sourced from publicly available datasets, and no proprietary or confidential data are used. Throughout the paper, every mention or use of these data sources has been properly and accurately cited. We strongly urge all users to adhere to the highest ethical standards when using our dataset, ensuring fairness, transparency, and responsibility in their research. Any usage of the dataset that may lead to harm or pose a detriment to society is strictly forbidden.

## ACKNOWLEDGMENTS

We would like to express our great gratitude to the anonymous reviewers for their kind comments. This work was supported by the National Natural Science Foundation of China (No. 62206246, No. NSFCU23B2055, No. NSFCU19B2027), the Fundamental Research Funds for the Central Universities (226-2023-00138), Yongjiang Talent Introduction Programme (2021A-156-G), CIPSC-SMP-Zhipu Large Model Cross-Disciplinary Fund, Ningbo Science and Technology Special Projects under Grant No. 2023Z212, Information Technology Center and State Key Lab of CAD&CG, Zhejiang University. We gratefully acknowledge the support of Zhejiang University Education Foundation Qizhen Scholar Foundation.

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

# A  APPENDIX

## A.1  SOURCE DATA INFORMATION

In order to facilitate a better understanding of our paper, here we provide a detailed exposition of the dataset utilized in our paper.

**Held-in Tasks**

- **ToolBench** (Qin et al., 2024). Toolbench is a function call dataset generated by selecting several APIs from an API library and synthesizing instructions using GPT-3.5. Due to the potentially disparate nature of the extracted APIs, many instructions within Toolbench may pose challenges in logical comprehension. To address this issue, we manually establish templates and filter them based on the complexity of function invocations. Additionally, there is another version of ToolBench called **StableToolBench** (Guo et al., 2024b) that utilizes GPT-4 and caching mechanisms to achieve stable API calls, which we use for end-to-end task evaluation.

- **ToolAlpaca** (Tang et al., 2023). The construction method of ToolAlpaca is akin to that of ToolBench. The quality of its instructions is relatively higher, yet most instructions can be executed within 1-3 function calls. We filter out instructions with only a single function call.

- **ALFWorld** (Shridhar et al., 2021). ALFWorld is a household dataset requiring the agent to navigate through the room and manipulate objects. It includes human-annotated gold trajectories, which we directly utilize to construct node chains and workflow graphs.

- **WebShop** (Yao et al., 2022). WebShop is an online shopping dataset in a website environment. We use the gold trajectories collected through GPT-4 in Song et al. (2024) to construct our benchmark.

- **OS** (Liu et al., 2024b). OS is an interaction dataset based on operating systems, where agents are required to complete operations through shell commands. We gather gold trajectories from AgentInstruct (Zeng et al., 2024a) to aid in the construction of our benchmark.

- **LUMOS** (Yin et al., 2024). LUMOS is a dataset that models agent planning-related datasets using a unified format. Within LUMOS, the OnePass planning data (LUMOS-O) aligns well with our workflow construction. We select the math, commonsense, and multimodal reasoning components as the foundation for building our reasoning tasks' workflow.

- **WikiHow** (Koupaee & Wang, 2018). WikiHow contains open-world planning tasks and complex linear process data. We collect and filter data based on task topics and directly construct graph workflows based on the process data.

**Held-out Tasks**

- **Seal-Tools** (Wu et al., 2024). Seal-Tools is a function call dataset obtained entirely through self-instruction using ChatGPT.

- **InterCodeSQL** (Yang et al., 2023a). InterCodeSQL is an embodied dataset that generates SQL instructions based on user intent and interacts with a graph database.

## A.2 HUMAN VERIFICATION

We divide the test data into five parts and invite five NLP volunteers to evaluate the quality of the node chains and workflow graphs based on the following principles:

1. **Granularity.** The decomposition of the node chain should meet the smallest executable granularity that can be derived from the task description and action list. This means that nodes should not combine subtasks (indicating granularity is too large) or introduce information that cannot be obtained from existing information (model self-association, indicating granularity is too small).

2. **Logic.** The workflow graph should follow logical sequencing that satisfies the execution relationships between nodes.

3. **Task Quality.** The tasks themselves should not exhibit any obvious quality issues.

We discard data that does not adhere to the above criteria and obtain the final test set.

## A.3 BENCHMARK STATISTICS

Figure 7 illustrates the statistics of our benchmark. Our training set comprises 18,679 instances, with data evenly distributed across four types: function call, problem-solving, embodied, and open-grounded. The test set consists of 2,146 instances, with 33.69% dedicated to held-out tasks to evaluate the generalization capability of the trained model. Figure 8 is the distribution of our benchmark based on the number of nodes in the workflow. The majority of the data is in the range of 2 to 10 steps, with a smaller portion falling within the 10 to 20 steps range. The average number of nodes across all data points is 4.17.

## A.4 ABLATION STUDY

Here we investigate what changes in the performance of the agents would occur if we provide the gold node chains to the agents and task it solely with predicting the dependencies between nodes (i.e., the edges of the workflow graph). The evaluation is still in a two-shot manner. The experimental results are shown in Table 5. We can observe a clear improvement in performance after providing the gold node chain. Giving the gold node chain can alleviate issues related to granularity and explicitness (as defined in Figure 4). However, predicting the graph relationships between nodes remains a challenging task, which contributes to an overall performance that is still subpar.

## A.5 TRAINING SETUPS

We also apply the LlamaFactory (Zheng et al., 2024b) framework to train the models. We fine-tune Qwen-2-7B and InternLM-2.5-7B with full parameters using DeepSpeed (Rasley et al., 2020). We

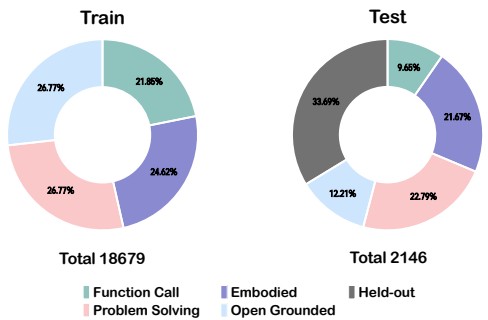
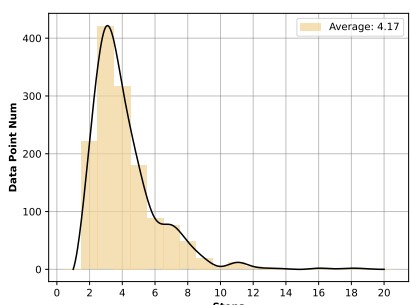

Figure 7: **Statistics of Our Benchmark.** We also include held-out tasks in the test set.

Figure 8: **Workflow Steps Distribution** on the whole benchmark.

Table 5: **Ablation Study.** We select some representative models to examine what will happen if we provide the gold node chains to the agents and task it solely with predicting the edges of the workflow graph. *w/o* represents traditional $f1_{\mathrm{graph}}$ without gold node chains and *w/* stands for generating edges with the aid of gold node chains.

| Model | 📚Function Call | | ✏️Problem-Solving | | 🏆Embodied | | 🔬Open-Grounded | | Average | |
|---|---|---|---|---|---|---|---|---|---|---|
| | w/o | w/ | w/o | w/ | w/o | w/ | w/o | w/ | w/o | w/ |
| Claude-3.5 | 55.06 | 72.75 | 55.50 | 89.27 | 56.71 | 78.80 | 42.88 | 62.06 | 52.53 | 75.72 |
| GPT-4 | 62.11 | 76.97 | 55.24 | 84.19 | 56.17 | 77.91 | 36.36 | 59.44 | 52.47 | 74.63 |
| Qwen-2-7B | 55.50 | 79.71 | 52.13 | 74.81 | 46.25 | 50.54 | 20.89 | 42.93 | 43.69 | 62.00 |
| InternLM-2.5-7B | 52.99 | 67.63 | 57.80 | 81.04 | 48.09 | 59.87 | 21.27 | 40.28 | 45.03 | 62.20 |
| Qwen-1.5-14B | 46.86 | 65.51 | 43.89 | 77.29 | 44.14 | 57.68 | 21.44 | 39.10 | 39.08 | 59.89 |
| Phi-3-medium | 47.26 | 65.80 | 54.85 | 80.98 | 49.99 | 52.21 | 23.77 | 33.33 | 43.97 | 58.08 |
| Llama-3.1-70B | 52.72 | 69.62 | 57.05 | 86.91 | 55.52 | 72.70 | 33.06 | 42.70 | 49.59 | 67.98 |
| Qwen-2-72B | 52.31 | 70.81 | 58.13 | 79.99 | 58.49 | 72.46 | 32.89 | 53.59 | 50.46 | 69.21 |

include a two-shot prompt in the input to enhance the model's generalization during both training and testing. Both models utilize identical hyperparameters. The detailed hyperparameter settings are outlined in Table 6. All the experiments are conducted on 3 NVIDIA 80GB A100 GPUs.

Table 6: Detailed training hyperparameters used in our paper.

| Name | Value |
|---|---|
| cutoff len | 4,096 |
| epochs | 3 |
| batch size | 12 |
| batch size per device | 2 |
| gradient accumulation steps | 2 |
| learning rate | 1e-5 |
| lr scheduler type | cosine |
| warmup ratio | 0.1 |
| bf16 | true |

## A.6 CASE STUDY OF O1

Below are two cases we test on OpenAI's current most powerful reasoning model, O1 (OpenAI, 2024). In the first case, O1 successfully identifies various subtasks but makes an error in predicting the dependency between node 1 and node 2/3 when generating the graph, failing to recognize their parallel relationship. In the second case, based on environmental priors, the agent needs to first locate the potato and then cool it using the fridge. Due to lacking this specific knowledge, O1 assumes the potato is already inside the fridge, thus missing the subtask of finding the potato. These observations confirm what has been mentioned in Valmeekam et al. (2024) that Large Reasoning Models (LRM) still can't plan. We analyze that the complexity of planning lies in the interaction with the environment, which requires a substantial amount of environmental commonsense knowledge.

Tasks like math reasoning or code generation only involve step-by-step deduction from the problem to the solution. Therefore, to enhance the planning ability of LLM agents, integrating world knowledge (Yu et al., 2024; Qiao et al., 2024a; Guetta et al., 2024) or world models (Dawid & LeCun, 2023; Hu & Shu, 2023; Wong et al., 2023) into the agent system remains crucial. This is also why a correct workflow can serve as prior knowledge to enhance agent planning.

---

**Case Study of O1 (a)**

**Task:** Please analyze the access_logs.txt file using the machine learning algorithm to detect any potential malicious activity. Afterwards, retrieve the network policy for the library with ID 'wscDOqa63Giq' regarding internet access. Finally, get the digital PR metrics for the 'Beauty Revolution' campaign that took place from '2022-01-01' to '2022-12-31' on platforms such as Twitter, Facebook, and Instagram.
**The tool list you can select from:** [tool_list]

**Gold Nodes and Edges**

**Node:**
1: Analyze access_logs.txt for potential malicious activity using machine learning.
2: Retrieve network policy for library 'wscDOqa63Giq' regarding internet access.
3: Get digital PR metrics for 'Beauty Revolution' campaign from '2022-01-01' to '2022-12-31' on Twitter, Facebook, and Instagram.
**Edge:**
(START, 1) (START, 2) (START, 3) (1, END) (2, END) (3, END)

**Predicted Nodes and Edges**

**Node:**
a. Analyze the access_logs.txt file for potential malicious activity using a machine learning algorithm.
b. Retrieve the network policy for the library with ID 'wscDOqa63Giq' regarding internet access.
c. Get the digital PR metrics for the 'Beauty Revolution' campaign from '2022-01-01' to '2022-12-31' on Twitter, Facebook, and Instagram.
**Edge:**
(START, 1) (1, 2) (1, 3) (2, END) (3, END)

---

**Case Study of O1 (b)**

**Task:** You are in the middle of a room. Looking quickly around you, you see a cabinet 20, a cabinet 19, a cabinet 18, a cabinet 17, a cabinet 16, a cabinet 15, a cabinet 14, a cabinet 13, a cabinet 12, a cabinet 11, a cabinet 10, a cabinet 9, a cabinet 8, a cabinet 7, a cabinet 6, a cabinet 5, a cabinet 4, a cabinet 3, a cabinet 2, a cabinet 1, a coffeemachine 1, a countertop 2, a countertop 1, a diningtable 2, a diningtable 1, a drawer 6, a drawer 5, a drawer 4, a drawer 3, a drawer 2, a drawer 1, a fridge 1, a garbagecan 1, a microwave 1, a sinkbasin 1, a stoveburner 4, a stoveburner 3, a stoveburner 2, a stoveburner 1, and a toaster 1. Your task is to: put a cool potato in garbagecan.
**The action list you can select from:**
1. go to recep
2. task obj from recep
3. put obj in/on recep
4. open recep
5. close recep
6. toggle obj recep
7. clean obj with recep
8. heat obj with recep
9. cool obj with recep
where obj and recep correspond to objects and receptacles.

**Gold Nodes and Edges**

**Node:**
1: go to where the potato is located
2: take potato from where it is located
3: go to fridge
4: cool potato with fridge
5: go to garbagecan
6: put potato in/on garbagecan.
**Edge:**
(START, 1) (1, 2) (2, 3) (3, 4) (4, 5) (5, 6) (6, END)

**Predicted Nodes and Edges**

**Node:**
Go to fridge 1
Take cool potato from fridge 1
Go to garbagecan 1
Put potato in garbagecan 1
**Edge:**
(START, 1) (1, 2) (2, 3) (3, 4) (4, END)

## A.7 PROMPT FOR BENCHMARK CONSTRUCTION

---

**Prompt for Node Chain Generation**

**Function Call Tasks**

Here is a multi-hop query and a segment of its solution trajectory in conversation format. Please summarize the content of the last step based on the "Action" results, and generate a node task. Node tasks should focus on the API related to the current action, and there should be no duplication between node tasks, make sure the task is clear, concise accurate, and specific.
Please note that the trace may contain some API call errors. Please ignore the errors and focus on the last Thought and Action step. Node task focuses more on planning rather than the specific API call or the results after execution so that specific API names should not appear. Here are two examples for you:
**Examples**
Now it's your turn.
Query: **Query**
Trajectory: **Trajectory Segment**

**Embodied Tasks**

I will provide you with an analysis of a successful trajectory of a task that interacts with the environment. Please identify the key factors that contribute to success. Based on this analysis, you need to generate workflow to help increase the success rate of future endeavors. The available actions are : **Action List**
Your output should be a sequence of subtasks that you would take to complete the task and in the format of: Workflow: A -> B -> C
Here are two examples for you:
**Examples**
Now it's your turn.
Query: **Query**
Trajectory: **Trajectory**

---

**Prompt for Workflow Graph Construction**

You are a planner who is good at task planning. Next, I will give you a task and some node of the subtasks. Some subtasks may not be dependent on each other, while others may have

> dependencies. Please convert these nodes of subtasks into a topology diagram based on the task relevance in the workflow. The Graph should start with the START node, and end with the END node.
> Here are two examples for you:
> **Examples**
> Now it's your turn.
> Task: **Task**
> Nodes: **Subtask Nodes**

## A.8 LIMITATIONS

This paper still has certain limitations that must be acknowledged: *a)* While we have enforced strict quality control on the node chain and workflow graph, it is inevitable that some queries themselves may have quality issues. The synthesis of complex queries remains an unresolved issue, which we leave for future work. *b)* All our data is collected from existing general domain datasets, inevitably missing scenarios that are not covered. For example, some tasks require heterogeneous actions (e.g. needing both function calls and embodied actions) to be completed. *c)* In addition to natural language, workflows can also be represented in code form (e.g. PDDL) (Zeng et al., 2023; Li et al., 2024c; Zuo et al., 2024), which we plan to incorporate in the future. *d)* Our workflow currently follows a one-pass generation paradigm. In the future, we plan to introduce an iterative paradigm where the workflow can be iteratively generated and evolve based on environmental feedback. *e)* Workflows in this paper assume that all nodes need to be traversed to complete the whole task. We do not cover some scenarios where choices, loops, or more complex structures (van der Aalst et al., 2003a) are in the graph.

## A.9 ANALYSIS OF THE TOPOLOGICAL ORDERS NUMBER IN NODE CHAIN EVALUATION

Due to the high time complexity of matching all the topological sorts of a graph, especially when there are many parallel nodes in the graph, we opt for a sampling approach to save time on matching. We analyze the distribution of the number of topological sorts for all workflow graphs in the test set, as shown in Table 7. It can be observed that the majority of workflows have a number of topological sorts within 10, and data exceeding 20 is quite sparse. Therefore, striking a balance between computational efficiency and evaluation accuracy, we chose 20 as the sampling quantity.

Table 7: The distribution of the number of topological sorts for all workflow graphs in the test set.

| **Topo Num** | $\leq 5$ | $\leq 10$ | $\leq 20$ | $\leq 50$ | $\leq 100$ |
|---|---|---|---|---|---|
| **Percentage** | 86.39% | 92.82% | 96.01% | 98.22% | 98.32% |

