# OpenReview forum: "Benchmarking Agentic Workflow Generation"
_ICLR.cc/2025/Conference — ICLR 2025 Poster_

### Official Review · Reviewer_nenD · 2024-10-20

**Soundness:** 3
**Presentation:** 3
**Contribution:** 3
**Rating:** 6
**Confidence:** 5

**Summary:**

This paper introduces WORFBENCH, a workflow generation benchmark designed to assess LLMs' ability to handle multi-faceted scenarios and complex graph-structured workflows. The experiments reveal significant gaps in LLM performance, especially in graph-based workflow prediction, which offer some interesting insights.

**Strengths:**

- 1. The scope of this paper makes sense, and evaluating the workflow is an important factor for agentic task solving.

- 2. Figure 2 clearly presents the main idea of the paper.

- 3. This benchmark includes some comprehensive topics of different agentic tasks.

**Weaknesses:**

- 1. A concern is that similar benchmarks have already been proposed but are not compared and mentioned in this paper, which may affect its overall soundness. For example, PlanBench [1], published two years earlier, covers similar scopes, and CAT-Bench [2], released four months before the ICLR submission, also overlaps in focus, etc.

[1] Valmeekam, Karthik, et al. "Planbench: An extensible benchmark for evaluating large language models on planning and reasoning about change." Advances in Neural Information Processing Systems 36 (2024).

[2] Lal, Yash Kumar, et al. "CaT-BENCH: Benchmarking Language Model Understanding of Causal and Temporal Dependencies in Plans." arXiv preprint arXiv:2406.15823 (2024).

- 2. It is unsure about the specific usage of the dataset. Since it is widely recognized that workflows are difficult to benchmark and planning can be dynamic, it is hard to determine if one decision path is truly superior to others as tasks become more complex.

- 3. There is a lack of discussion on whether the datasets remain reliable as the number of planning steps increases. Does the generated workflow still prove useful in such cases? This might introduce significant hallucinations, potentially harming task-solving effectiveness.

- 4. The results in Table 3 are not convincing. If adding workflows (+W) improves performance so significantly, wouldn't it be logical to have these models first generate the workflow and then run the experiments (for many previous projects or papers)?

**Questions:**

- 1. It seems that GPTSwarm [3] and TextGrad [4] offer an effective solution for DAG-based task solving, improving workflows in graph-based settings. It would also be helpful to discuss related work, and suggest supplementing at least 3 related papers with earlier studies on workflow generation or graph-based agentic systems like [3], [4].

[3] Zhuge, Mingchen, et al. "GPTSwarm: Language Agents as Optimizable Graphs." Forty-first International Conference on Machine Learning.

[4] Yuksekgonul, Mert, et al. "TextGrad: Automatic" Differentiation" via Text." arXiv preprint arXiv:2406.07496 (2024).

- 2. The difficulty of this task is not very challenging. I suggest adding (or splitting) a hard set of tasks that are visibly more difficult and showing the results.

- 3. To be honest, Q1 in the paper did not provide much insight. It seems to this point that is more like common sense and doesn’t require much space to verify.

---

> ### Comment · Reviewer_nenD · 2024-11-16
> **thanks**
>
> Thanks to the authors. While I still have some concerns that prevent me from fully supporting the strong acceptance of this paper, the authors have acknowledged that some of these concerns are indeed limitations. I have no further questions and will maintain my original score.

---

### Official Review · Reviewer_EjZY · 2024-11-03

**Soundness:** 3
**Presentation:** 3
**Contribution:** 3
**Rating:** 8
**Confidence:** 3

**Summary:**

This paper proposes extending the evaluation of workflows in agentic workflow contexts from purely sequential workflows to a slightly richer graph structure that include parallelism, represented as DAGs.

I think this is an important step, but also just a first step into much needed generalization from agentic workflows to real-work scenarios that often involve much more complex graph structures. The area of analysis workflow analysis and optimization of workflows has existed as a rich and mature area of study in the research fields of business process management and process mining [1,2], and decades of work has been done in those fields to study various representations of workflows. One difference is that those communities originally focused on representations of workflows that represent business processes and/or human execution of work, while the focus here is on workflow representations of LLM execution steps. I believe that there is no meaningful difference between the two from a workflow perspective.

I applaud this paper for taking a first step in the direction of bringing those fields closer together, as there seems to be much to gain.

**References**:

[1] van der Aalst, W. M. P., Van Dongen, B. F., Herbst, J., Maruster, L., Schimm, G., & Weijters, A. J. (2003). Workflow mining: A survey of issues and approaches. Data & knowledge engineering, 47(2), 237-267.

[2] van der Aalst, W. M. P. (2016). Data science in action (pp. 3-23). Springer Berlin Heidelberg.

**Strengths:**

The paper distinguishes gaps between sequence planning capabilities and graph planning capabilities of LLM agents that were previously unknown. This is an important finding.

More importantly, I see great value in the steps that this paper has taken (even if limited) to extend (benchmarking of) agentic workflows from purely linear workflows to slightly richer workflows that also cover parallel execution. This is a small step towards integrating decades of progress of workflow analysis from areas outside of the LLM and machine learning communities into agentic workflows.

**Weaknesses:**

- The paper chooses to formalize workflows as DAGs. This representation is able to capture both sequential execution as well as parallel execution, but it still is unable to capture many other relevant structures that play a role in real-life scenarios. A DAG is for example unable to represent a decision point (choice), and unable to represent loops (repeated execution). I refer to [3] for foundational work on workflow patterns that contains a rich collection of workflow patterns, of which only patterns 1, 2, and 3 are captured in the current representation.

- Modern representations of workflows in the business process management community include so-called workflow-nets [4] (a subclass of Petri nets [5]), and BPMN [6]. Note that the DAG formalism proposed in this paper is also equivalent to a subclass of Petri nets, namely the class of so-called “marked graphs” [7]. While it seems natural to study prior literature in workflow analysis outside of the LLM context when extending (evaluation of) agentic workflow to rich workflow patterns, this seems to be lacking, which is a missed opportunity. I encourage authors to at least incorporate some of these works into their related work section.

- Minor: Figure 2 is small and hard to read.

**References**:

[3] van der Aalst, W. M.P., Ter Hofstede, A. H., Kiepuszewski, B., & Barros, A. P. (2003). Workflow patterns. Distributed and parallel databases, 14, 5-51.

[4] van der Aalst, W. M.P. (1997). Verification of workflow nets. In: International Conference on Application and Theory of Petri nets (pp. 407-426). Springer.

[5] Peterson, J. L. (1977). Petri nets. ACM Computing Surveys (CSUR), 9(3), 223-252.

[6] Dijkman, R. M., Dumas, M., & Ouyang, C. (2008). Semantics and analysis of business process models in BPMN. Information and Software technology, 50(12), 1281-1294.

[7] Commoner, F., Holt, A. W., Even, S., & Pnueli, A. (1971). Marked directed graphs. Journal of Computer and System Sciences, 5(5), 511-523.

**Questions:**

- In WorkFEval, nodes are represented using sequence-BERT, where it is checked if the similarity of the predicted and the ground truth node *at the same index* is above some similarity threshold. I wonder how this works within a parallel block. Take for example the Graph Workflow of Figure 1 as example: nodes 1 and 2 are in parallel and if the ground truth workflow and we also have a predicted workflow that has two parallel nodes at the start, then how do we know which predicted node to similarity-match to which node from the ground truth graph?
- There is over a decade of work in workflow graph similarity metrics in the business process management literature. See e.g. [8] for a survey. Have authors considered using any of those existing methods out-of-the-box for comparing the ground truth workflow graph to the predicted workflow graph?
- Comparing the node chain to the gold workflow graph: it seems that this problem can just be reduced to the reachability problem in Petri nets. Since the chosen DAG representation is just a marked-graph, there is a well-known result from Petri net theory that reachability in marked graphs is solvable in polynomial time [9]. The current implementation decision of generating all possible topological sequences seems exponential in time (in the case where all nodes are parallel). This may perhaps be OK with the size of DAGs that we practically encounter today. I wonder if authors envision that in the future we could be dealing with DAGs that are large enough that this may become a limitation? (I believe that the alignments algorithm can provide a more efficient comparison of node chain to workflow graph and also provide more principled metrics for this problem (see [10, 11]).

**References:**

[8] Schoknecht, A., Thaler, T., Fettke, P., Oberweis, A., & Laue, R. (2017). Similarity of business process models — a state-of-the-art analysis. ACM Computing Surveys (CSUR), 50(4), 1-33.

[9] Esparza, J., & Nielsen, M. (1994). Decidability issues for Petri nets. Petri nets newsletter, 94, 5-23.

[10] van der Aalst, W. M. P. , Adriansyah, A., & van Dongen, B. (2012). Replaying history on process models for conformance checking and performance analysis. Wiley Interdisciplinary Reviews: Data Mining and Knowledge Discovery, 2(2), 182-192.

[11] van Dongen, B. F. (2018). Efficiently computing alignments: using the extended marking equation. In Business Process Management: 16th International Conference, BPM 2018, Sydney, NSW, Australia, September 9–14, 2018, Proceedings 16 (pp. 197-214). Springer International Publishing.

---

### Official Review · Reviewer_Dn7v · 2024-11-08

**Soundness:** 3
**Presentation:** 3
**Contribution:** 3
**Rating:** 6
**Confidence:** 3

**Summary:**

The paper presents WORFBENCH, a benchmark designed to evaluate LLM-generated workflows in complex reasoning and planning tasks. It introduces subsequence and subgraph matching algorithms for assessing workflows across diverse and intricate planning scenarios. Experiments highlight a notable performance gap between linear and graph-based workflows, with even GPT-4 showing a 15% deficiency in graph planning. It also reveals that well-structured workflows can improve downstream task performance and inference efficiency.

**Strengths:**

- The writing and presentation is good.
- The motivation of standardizing the evaluation of agent workflow is impressive.
- The evaluation is extensive and the insights from the evaluation are helpful.

**Weaknesses:**

N/A

**Questions:**

This paper introduces WORFBENCH, a benchmark designed to evaluate LLM-generated workflows across diverse scenarios and complex graph structures, utilizing subsequence and subgraph matching algorithms for precise assessment. The evaluation is comprehensive, offering valuable insights into how structured agent workflows can enhance LLM planning capabilities. This work makes a significant contribution to the agent community. However, regarding insights of the graph-based agent workflows, I have several questions.
- In the evaluation, the authors consider different actions as nodes across scenarios such as function-calling and embodied tasks. Have you considered more complex scenarios with heterogeneous action nodes, where some nodes represent function calls while others represent embodied actions? An evaluation or discussion on this point could be interesting.
- More information on the training process to embed structured workflows as knowledge in LLMs would be helpful. For example, is there any insight of which training method can be more beneficial to training the LLMs with well-structured graph workflows, such as instruction-tuning or reinforcement learning through self-exploration?

---

### Official Review · Reviewer_eFu6 · 2024-11-08

**Soundness:** 3
**Presentation:** 3
**Contribution:** 3
**Rating:** 6
**Confidence:** 4

**Summary:**

This paper proposed a new benchmark to evaluate the abilities of LLMs in generating agentic workflows.
The insight of this paper lies in that the agentic workflows can be considered as a directed acyclic graph and therefore the generation of the workflow can be formulated as the generation of the nodes and edges in the graph.
The evaluation is done by measuring the similarity of the model generated workflow graph to the ground graph.
The experimental results across multiple models demonstrate that current models cannot handle the planning on the workflow graph level very well, and the most advanced models are still not performing on a good level on this benchmark.

**Strengths:**

1. The idea of evaluating agentic workflow generation is interesting and novel.
2. The paper puts in the work to evaluate a wide range of models.
3. The evaluated scenarios cover a wide range of agentic use cases.

**Weaknesses:**

1. The matching between the model generated nodes and the ground truth nodes are done by using sentence bert. This may introduce errors in the evaluation step if the matching is not correct.
2. The evaluation metrics mostly focus on the similarity to a ground truth graph, but not on how the generated workflows can complete the task correctly. There might be more than one graph than can complete the given task.

**Questions:**

1. What if the model is given the ground truth nodes and they only need to predict the edges between the nodes? Would this improve the performance?
2. Does the benchmark contains tasks that can be solved by using multiple different workflow graphs? Can the current evaluation metrics account for it?

---

### Official Review · Reviewer_jY5E · 2024-11-10

**Soundness:** 2
**Presentation:** 2
**Contribution:** 2
**Rating:** 6
**Confidence:** 4

**Summary:**

This paper introduces WORFBENCH, a unified benchmark designed to evaluate the workflow generation capabilities of large language models (LLMs) across diverse scenarios and complex graph-based workflow structures. It also presents WORFEVAL, a evaluation protocol using subsequence and subgraph matching algorithms for accurate assessment of workflow generation is proposed. Author claimed following key contributions: 1)new features (Multi-faceted Scenarios and Complex Workflow Structures, Strict Quality Control); 2)WorFEval Evaluation (utilizes advanced matching algorithms to assess LLM performance on both linear and graph-based workflows quantitatively); 3)Comprehensive Experiments

**Strengths:**

S1)The problem highlighted by the paper is valid and emerging
S2)The dataset has some interesting features
S3)The experiments seem to be extensive

**Weaknesses:**

W1)the evaluation scores f1_chain and f1_graph in Section 2.4 were introduced as the measures for all evaluations in Section. But they are given without solid foundation why they are formulated and the right measures for the workflow chain/graph.
W2) Quality control protocol is very subjective and manual, it’s difficult to judge the quality of data of the benchmark
W3) Many technical details are not very clear (see questions)

**Questions:**

Q1)Is the training code included in the supplemental material? If yes, where to find and run it
Q2)The paper provides some number such as ”1k training samples, 2146 test samples..” → how significant of these numbers towards the task and state of the art?
Q3)How are the gold nodes and edges checked for correctness?

-------------------
Some questions were answered by authors, so, I raised the score to 6

---

### Meta-Review · Area_Chair_KXe1 · 2024-12-21

**Metareview:**

This paper introduces WorFBench, a benchmark designed to evaluate the workflow generation capabilities of LLMs in complex reasoning and planning tasks. The authors also develop WorFEval, an evaluation protocol that utilizes subsequence and subgraph matching algorithms to quantitatively assess the quality of generated workflows.

Overall, the reviewers expressed strong appreciation for the paper, acknowledging its novelty, significance, and timeliness; additionally, they believe the experiments part are well conducted and reveal some insightful findings. But meanwhile, some concerns are raised, mainly about 1) the lack of a solid justification for the proposed evaluation metrics; 2) some related works like PlanBench are not discussed; 3) certain technical details are not clearly presented; 4) the formalization of workflows as DAGs could bring in limitations in representing other important workflow constructs; and 5) it is unclear whether the proposed framework can effectively generalize to more difficult and complex tasks.

The authors' responses to these concerns were engaging and effectively addressed the majority of the reviewers' issues. As a result, all reviewers unanimously voted to accept the paper. The AC concurs with this decision and believes the work will generate significant interest within the ICLR community.

**Additional Comments On Reviewer Discussion:**

The major concerns raised by reviewers are listed in my meta-review. Overall, most of these concerns have been carefully discussed, and the reviewers have expressed satisfaction with the responses. One unacknowledged concern is regarding 1) the lack of a solid justification for the proposed evaluation metrics. The AC has read the follow-up message from the author and believes this is reasonably addressed. Additionally, the reviewer who raised this concern still rated this paper as 6 (i.e., above the acceptance threshold), which leads the AC to consider this concern as minor.

Given these points, the AC concludes that no major concerns remain after the discussion period and recommends accepting the paper.

---

### Decision · Program_Chairs · 2025-01-22

Accept (Poster)